# Time-Restricted Eating: A Novel and Simple Dietary Intervention for Primary and Secondary Prevention of Breast Cancer and Cardiovascular Disease

**DOI:** 10.3390/nu13103476

**Published:** 2021-09-30

**Authors:** Rebecca A. G. Christensen, Amy A. Kirkham

**Affiliations:** 1Dalla Lana School of Public Health, University of Toronto, Toronto, ON M5T 3M7, Canada; r.christensen@mail.utoronto.ca; 2Faculty of Kinesiology and Physical Education, University of Toronto, Toronto, ON M5S 2C9, Canada

**Keywords:** breast cancer, cardiovascular disease, time-restricted eating, time-restricted feeding, intermittent fasting, metabolic syndrome, fasting

## Abstract

There is substantial overlap in risk factors for the pathogenesis and progression of breast cancer (BC) and cardiovascular disease (CVD), including obesity, metabolic disturbances, and chronic inflammation. These unifying features remain prevalent after a BC diagnosis and are exacerbated by BC treatment, resulting in elevated CVD risk among survivors. Thus, therapies that target these risk factors or mechanisms are likely to be effective for the prevention or progression of both conditions. In this narrative review, we propose time-restricted eating (TRE) as a simple lifestyle therapy to address many upstream causative factors associated with both BC and CVD. TRE is simple dietary strategy that typically involves the consumption of ad libitum energy intake within 8 h, followed by a 16-h fast. We describe the feasibility and safety of TRE and the available evidence for the impact of TRE on metabolic, cardiovascular, and cancer-specific health benefits. We also highlight potential solutions for overcoming barriers to adoption and adherence and areas requiring future research. In composite, we make the case for the use of TRE as a novel, safe, and feasible intervention for primary and secondary BC prevention, as well as tertiary prevention as it relates to CVD in BC survivors.

## 1. Introduction

Breast cancer is the most common malignancy among women worldwide, with 1 in 8 North American women expected to be diagnosed in their lifetime [1]. While there is no single biological target for the primary prevention of breast cancer, diet, adult weight gain, and obesity are estimated to be responsible for up to 50% of cases [2,3,4,5]. Metabolic dysfunction, signified by presence of hyperglycemia, dyslipidemia, hypertension, and abdominal obesity, is primary driver in the risk of type 2 diabetes and cardiovascular diseases [6]. However, in the past decade, metabolic dysfunction has also emerged as an underlying determinant of the relationship between obesity and breast cancer risk [7,8,9,10]. Other systemic factors associated with overweight/obesity and cardiovascular disease, such as chronic inflammation [11] and oxidative stress [12], are also associated with breast cancer risk [13,14,15,16,17]. Thus, strategies for the primary prevention of breast cancer are also likely to impact the risk of cardiovascular and metabolic diseases. 

Fortunately for those who receive a breast cancer diagnosis, the death rate for early stage (non-metastatic) breast cancer has dropped by over 40% in the last 40 years [18]. Concomitant to improved cancer survival, the death rate due to cardiovascular disease has increased and now approaches the rate of cancer death [19]. In fact, women diagnosed with breast cancer are at a 2 to 3-fold elevated risk of cardiovascular-related death relative to the general population perpetually after their diagnosis [19]. While the elevated cardiovascular risk partially arises from the presence of pre-existing shared risk factors, it is also compounded by breast cancer treatments that result in direct toxicity to the heart (‘cardiotoxicity’), metabolic dysfunction, as well as lifestyle toxicity (i.e., physical inactivity, poor diet, weight gain [20,21]). The biological and behavioral sequelae resulting from breast cancer treatment can persist long into the survivorship period, as evidenced by the excess risk of cardiovascular death increasing to >5-fold at 10+ years after diagnosis [19]. In the 20 years following a breast cancer diagnosis, the risk of recurrence of breast cancer ranges from 10–38% depending on diagnosis characteristics [22]. Similar to primary prevention of breast cancer, risk factor targets for secondary prevention as it relates specifically to prevention of cancer recurrence, second cancers, and cancer mortality include poor diet [23], physical inactivity [24] and obesity [25,26]. Thus, after a breast cancer diagnosis, lifestyle intervention strategies for secondary prevention will also have overlap with tertiary prevention as it relates to addressing the cardiovascular and metabolic sequelae of treatment and prevention of the related diseases. 

The multi-faceted and intertwined risk profile for breast cancer and cardiovascular disease thus creates a shared opportunity to reduce the risk of both conditions by treating their shared underlying biological and behavioral mechanisms. In this context, therapies that target multiple possible biologic and behavioral mechanisms or pathways for these conditions will be the most effective for prevention. 

Intermittent fasting is a relatively new dietary intervention that has recently gained substantial public interest. There are multiple different formats including alternative day fasting (i.e., restriction of caloric intake on 2 to 5 days/week alternated with ad libitum consumption), periodic prolonged fasting (i.e., 24 h to one week), and repeated daily fasting. The latter, referred to as Time Restricted Feeding (for animals) or Time-Restricted Eating (TRE) (for humans), involves the consumption of ad libitum energy intake within a set time window, ranging from 4–10 h, but most commonly 8 h. This is followed by a water-only fast for the remaining time in the 24-h period, typically 16 h (i.e., “16:8 TRE”). As 16:8 TRE requires relatively minor lifestyle changes and has simple instructions [27], it may be feasible as a long-term lifestyle intervention. Importantly, TRE has numerous health benefits that are relevant to the primary, secondary, and tertiary prevention of breast cancer. The purpose of this narrative review is to describe all relevant peer reviewed literature on the potential for TRE as a therapy for primary and secondary prevention of breast cancer, secondary prevention as it relates to recurrence or cancer mortality, as well as tertiary prevention as it relates to cardiovascular disease in breast cancer survivors. 

## 2. TRE Health Benefits and Mechanisms

### 2.1. Body and Fat Mass

Body mass is an important determinant of metabolic health, which is often signified by the absence of elevated cholesterol, triglycerides, blood pressure, blood sugar and waist circumference. Body mass index (BMI) is most often used to classify the level of health risk associated with body mass relative to height. Notably, having even a modest increase in body mass, such that BMI exceeds 25 kg/m^2^, as defined by having overweight, is a predictor of postmenopausal breast cancer [28], breast cancer mortality [29] and cardiovascular disease in women [30]. Accumulation of fat mass in particular is the driver of increased risk of breast cancer in postmenopausal women, as this is the main source of estrogen in the body after menopause [31]. Likewise for risk of cardiovascular disease, body fat is a stronger predictor than the more generalized measure of BMI [32]. In addition, weight gain is common after a breast cancer diagnosis, with 50–96% of women reporting weight gain during breast cancer treatment [33]. While weight gain is typically associated with an increase in both fat mass and fat-free mass, following a breast cancer diagnosis, the more common pattern observed is an increase in fat mass, but a decrease in fat-free mass [33,34]. 

Preclinical data have highlighted the significance of the timing of food intake in weight gain [35,36]. The circadian rhythm is the oscillation of physiological rhythms between activity and rest, feeding and fasting, nutrient utilization and storage drive by the natural day/night cycle [37]. The brain and nearly every peripheral organ have circadian timekeeping mechanisms that impact their function [38]. Late dinner times, late-night snacking, or eating during the night (as is common in night or shift workers) can chronically disrupt circadian rhythms, impacting gene expression and lead to metabolic dysfunction and disease including obesity [37]. Shortening the length of the eating window and skewing intake away from the evening/night helps to re-align food intake with the natural 24-h human cycle of feeding and fasting [39]. TRE thereby resets the body’s peripheral clocks, which results in improved oscillations in gene expression and enhanced energy metabolism [37]. 

A 16:8 ad libitum TRE protocol typically results in 20–30% spontaneous caloric restriction and mild weight loss of 1–4% over 1–12 weeks without the need to count calories [40]. A recent meta-analysis of 12 TRE intervention studies lasting 4–12 weeks with 294 participants reported an overall significant weight reduction of −0.9 kg (95% CI, −1.71 to −0.10) [41]. In a subgroup analysis of five studies including patients with metabolic abnormalities (i.e., overweight/obesity, pre-diabetes, metabolic syndrome), the weight loss was greater (−3.19 kg, 95% CI, −4.62 to −1.77) [41]. Longer duration studies of TRE are needed to determine if the amount of weight lost induced by TRE can meet or exceed the commonly accepted clinically relevant threshold of 5% of baseline weight [42,43]. The same meta-analysis also reported significant reductions in fat mass with TRE (−1.58 kg, 95% CI, −2.64 to −0.51) while preserving fat-free mass (−0.24 kg, 95% CI, −1.15 to 0.67) measured by bioelectrical impedance or dual-energy X-ray absorptiometry [41]. This is an important finding for the long-term viability of TRE as a health intervention, because weight loss interventions typically result in concomitant decreases in both fat and fat-free mass [44]. 

Preliminary evidence suggests that TRE reduces fat in ‘ectopic’ regions of the body (e.g., visceral, liver, intermuscular) [38,45,46,47]. Relative to whole-body fat mass or BMI, these specific locations of fat deposition are much more strongly linked to the primary risk of cardiovascular disease and breast cancer [48,49], and the risk of cardiovascular, breast cancer, and all-cause mortality among breast cancer survivors [50,51,52]. Given that breast cancer therapies, including chemotherapy, targeted therapy, and hormonal therapy, have been shown to result in rapid and persistent accumulation of visceral, liver, and intermuscular fat [53,54], TRE may be a promising therapy to employ during active treatment to prevent this metabolic toxicity. While there is interest in the use of intermittent fasting during chemotherapy treatment for breast cancer, the strategies employed to-date have involved longer periods of fasting (24–72 h), which are safe but potentially not widely feasible among humans [55]. The approach of a shortened window for eating each day with TRE may be more palatable for patients and may still provide protective effects against treatment toxicity including ectopic fat accumulation. The use of intermittent fasting during active treatment may also be effective for secondary prevention of cancer based on preliminary findings that nutrient deprivation sensitizes cancer cells to the damaging effects of chemotherapy [56,57].

### 2.2. Oxidative Stress and Inflammation

Oxidative stress and chronic inflammation are unifying features in the pathogenesis and progression in both cancer and cardiovascular disease and their shared risk factor of obesity [58]. Oxidative stress is a disrupted balance between the production of damaging reactive oxygen species and antioxidant defenses [59]. Accumulating evidence implicates the DNA damage and mutations of tumor suppressor genes associated with oxidative stress and reactive oxygen species as critical initial events in carcinogenesisis [60]. One study reported a relationship between higher levels of oxidative stress, as assessed by plasma lipoperoxides, and a two-fold greater risk of breast cancer recurrence (relative risk = 2.10, 95% CI 1.10–4.00) [17]. Elevated oxidative stress is also implicated in various types of cardiovascular disease, primarily through effects on endothelial function and myocardial calcium handling, which contribute to hypertension and/or arrhythmia [61]. Preliminary evidence suggests that TRE may reduce oxidative stress in men with pre-diabetes [62] and healthy men [63], but more evidence is needed in women with chronic disease. Notably, while excess body mass is associated with oxidative stress, TRE-induced weight loss was not a requisite for reduced oxidative stress in prediabetic men [62]. 

Obesity is said to be a state of chronic inflammation. Chronic inflammation promotes malignant transformation of cells, carcinogenesis, and progression and is a precursor to stroke, as well as mediates all stages of atherosclerosis and other cardiovascular disease events [58,64]. One pilot study of a 4-week 16:8 TRE intervention observed no significant changes in c-reactive protein [65], a marker of systemic inflammation. Conversely, two other studies employing longer durations of daily fasts (14–15 h) or intervention length (8 weeks) have observed significant decreases in the proinflammatory markers interleukin (IL)-6 and IL-1β [66,67]. Further, the reduction in proinflammatory markers was independent of weight loss in one study [66]. More research is needed to better elucidate the dose response effects of TRE on inflammation.

### 2.3. Metabolic Syndrome

Metabolic syndrome is a constellation of metabolic disturbances including hyperglycemia, dyslipidemia, hypertension, and abdominal obesity, that increase the risk of heart disease, stroke, and type 2 diabetes. Metabolic syndrome also appears to play an important role in breast cancer. A recent meta-analysis showed that women with metabolic syndrome have a 52% increased risk of breast cancer [68]. Among women diagnosed with breast cancer, metabolic syndrome increases the risk of recurrence or distant metastases and breast cancer mortality [69,70].

Glucose metabolism specifically has also recently emerged as a key biological mechanism in breast cancer development [71]. Biologic mechanisms underpinning this relationship include glucose-mediated upregulation of oncogenic pathways in non-malignant breast cells [72] and insulin resistance-related promotion of cellular proliferation and inhibition of apoptosis [73]. A meta-analysis reported that the clinical manifestation of impaired glucose control, type 2 diabetes, increases the risk of breast cancer by 23% [74]. The diagnostic blood marker for the diagnosis of diabetes, hemoglobin A1c, which provides a measure of the average blood glucose concentration over the previous 8–12 weeks, is associated with risk of breast cancer independent of diabetes [75], and the risk of cardiovascular disease in women without diabetes [76]. 

Two prospective cohort studies illustrate the potential for TRE to improve chronic glucose control as a strategy for primary and tertiary prevention of breast cancer. Among 2212 women with elevated BMI, each 3-h increase in habitual overnight fast time was associated with 19% lower odds of elevated hemoglobin A1c [77]. Among 2413 breast cancer survivors without diabetes, habitual overnight fasting duration was inversely associated with hemoglobin A1c [78]. Few TRE intervention studies have measured hemoglobin A1c, likely because the length of most interventions to-date (≤12 weeks) are not long enough to impact this chronic marker. However, one study with a 10-h eating window in patients with metabolic syndrome reported that hemoglobin A1c significantly decreased among those with elevated baseline levels ≥5.7% without a concurrent change in physical activity [79]. Further, a meta-analysis of 10 TRE intervention studies with 238 participants reported a statistically significant but modest reduction in fasting blood glucose (−2.96 mg/dL, 95% CI, −5.60 to −0.33) [41].

The individual components of metabolic syndrome are also linked to both breast cancer and cardiovascular disease. Hypertension is one of the main causal risk factors related to cardiovascular disease [80] with a strong, positive dose–response relationship with the risk of death from ischemic heart disease and stroke [81]. Breast cancer risk is also associated with hypertension, with several meta-analyses reporting a 7–38% higher risk of breast cancer among women with hypertension compared to normotensive women [82,83]. A meta-analysis of six TRE studies with 97 participants found modest but clinically significant decreases in systolic (−3.07 mmHg, 95% CI, −5.76 to −0.37) and diastolic (−1.77 mmHg, 95% CI, −4.51 to 1.07) blood pressure [41]. Importantly TRE may reduce blood pressure independent of weight loss [40]. 

Dyslipidemia, defined as elevated total or low-density lipoprotein (LDL) cholesterol, or low high-density lipoprotein (HDL) cholesterol, is another important metabolic risk factor. Research regarding the association between blood lipid levels and breast cancer incidence is mixed. Some studies have suggested an inverse relationship between lipid levels and breast cancer risk [84,85] while others have shown a positive association [86,87]. This discrepancy may be explained by inclusion of women taking cholesterol-lowering drugs (statins), as these treatments have been shown to reduce breast cancer incidence, recurrence, and mortality [88]. While there are still some discordant results, research tends to suggest that the level of HDL is inversely associated with breast cancer risk [89,90]. In contrast, the detrimental effect of elevated cholesterol on cardiovascular disease risk is well established [91]. A meta-analysis of 14 TRE studies with 343 participants reported significant reductions in triglycerides (−11.60 mg/dL, 95% CI, −23.30 to −0.27), but highly variable effects on LDL (0.05 mg/dL, 95% CI, −4.77 to 4.87) and HDL (1.01 mg/dL, 95% CI, −1.52 to 3.55). It is possible that favorable changes to LDL would be evident once clinically significant weight loss (>5% from baseline) is attained with longer adherence to TRE [40]. A number of shorter duration (1–8 weeks) TRE studies have reported significantly increased HDL levels, while others have not, potentially related to concomitant changes in metabolism that require further study [40]. While TRE may result in favorable changes to some aspects of the lipid profile, larger sample sizes and studies with longer TRE intervention duration are required to confirm these effects. 

The final component of the metabolic syndrome, abdominal obesity, is measured by elevated waist circumference (≥88 cm for women), a simple and practical anthropometric measure. The primary driver of the relationship between abdominal obesity and poor metabolic health is the volume of visceral fat. As discussed earlier, visceral adiposity is strongly linked to the development of cardiovascular disease and breast cancer [48,49] and related mortality [50,51,52]. Three studies employing 8–10-h eating windows for 12 weeks among individuals with obesity or metabolic syndrome reported a statistically significant decrease or trend in measures of visceral fat via dual-energy X-ray absorptiometry [45,47], and bioelectrical impedance [79]. Given these positive preliminary findings and the importance of this outcome, the effect of TRE on visceral fat merits further study.

### 2.4. Auxiliary Health Behavior Benefits

Physical activity is an important protective factor for breast cancer incidence [92,93], breast cancer mortality [94] and cardiovascular disease incidence and mortality [95] (including in breast cancer survivors) [96] Therefore, a concomitant reduction in physical activity, as has been known to occur with participation in a moderate or severe calorie restriction diet [97], could attenuate or mute the benefits of calorie restriction on prevention of breast cancer or cardiovascular disease. While evidence is preliminary, TRE does not appear to alter physical activity levels [45,47,98,99]. In fact, two studies have found that a TRE intervention without physical activity may modestly improve physical function in middle-aged and older adults [100,101]. Improved physical function could have downstream effects of increasing habitual physical activity, but this requires longer duration studies in targeted populations with poor physical function.

When TRE or other forms of intermittent fasting or caloric restriction are combined with a purposeful exercise training intervention, additive or synergistic favorable effects on a number of health outcomes relevant to breast cancer and cardiovascular disease have been reported, including cardiorespiratory fitness, body composition, fasting insulin and glucose, and insulin-like growth factor-1 [102,103,104,105,106]. For example, the addition of 16:8 TRE to a structured resistance training intervention in healthy women resulted in a significantly greater reduction in fat mass and percent body fat than resistance training alone and a similar gain in fat-free mass [107]. Therefore, in populations who are physically able, a combined intervention of TRE and exercise training may provide enhanced benefits for the primary, secondary, and tertiary prevention of breast cancer.

Shortening the eating window to follow TRE may incidentally result in dietary behavior changes that are independently linked to breast cancer incidence including reduced caloric intake as already discussed, as well as reductions in alcohol consumption and late-night snacking on sweet [2,3,108,109,110]. Among 99 healthy individuals or those with obesity, self-reported sleep quality but not duration was improved after following 16:8 TRE for 12 weeks [111]. 

### 2.5. Cancer-Specific Biological Effects

Disruption of circadian rhythms can be associated with abnormal cellular division associated with tumorigenesis [112]. Disruptions to the circadian rhythm are linked to breast cancer development through altered expression of circadian genes in the breast tissue in addition to the associated impaired glucose metabolism discussed earlier. Circadian clocks in the breast regulate the expression of numerous genes, and when disrupted can alter breast biology and promote cancer [112]. The potential link between shift work and risk of breast cancer illustrates this relationship. Women who have long-term exposure to rotating night and day work shifts, such as nurses or doctors, may have an increased risk of breast cancer [113]. Re-aligning the circadian clocks will result in improved oscillations in gene expression and enhanced energy metabolism. TRE may help to accomplish this through re-establishing the oscillations in the feeding-fasting cycle, but the effects of TRE have not been studied in shift workers who are also exposed to severe disruptions to oscillations in the sleep-wake cycle.

The regular exposure to a fasting period induced by TRE also has benefits for cellular health. Regular fasting activates cell signaling pathways and integrated adaptive responses between and within organs that increase the expression of antioxidant defenses, DNA repair, protein quality control, mitochondrial biogenesis, autophagy, and reduces inflammation [39]. This adaptive response confers resistance to oxidative and metabolic stress and the removal/repair of damaged molecules [39]. Through these mechanisms, TRE has the potential to modify biological mechanisms in common for a wide range of chronic disorders including, cancer, cardiovascular disease, diabetes, and neurodegenerative disorders [39]. Specific to cancer, there is evidence that repeated fasting can reduce cell proliferation, cancer progression, and metastases [114]. 

One compelling finding directly relevant to breast cancer is that breast cancer survivors without diabetes who reported habitually fasting overnight for less than <13 h, had a 36% higher risk of breast recurrence (local, regional, or distant recurrence, or new primary) (hazard ratio, 1.36, 95% CI, 1.05 to 1.76) [78]. In this prospective cohort study that followed 2413 breast cancer survivors for 7.3 years, [78] there were trends toward lower hazard of breast cancer-specific mortality (hazard ratio, 1.21, 95% CI, 0.91 to 1.60) and all-cause mortality (hazard ratio, 1.22, 95% CI, 0.95 to1.56) as well [78]. In addition to the total length of the nightly fasting duration, eating after 8 pm appeared to be a potential determinant of risk, as it was associated with increased chronic inflammation and higher BMI among these breast cancer survivors [78].

### 2.6. Heart Failure

There is preliminary evidence that TRE may be effective in primary and secondary prevention of heart failure. An observational study of 2001 patients undergoing cardiac catheterization without prior myocardial infarction or heart failure reported that prolonged nightly fasting or religious fasting was associated with a 71% reduced incidence of heart failure (adjusted hazard ratio, =0.29, 95% CI, 0.11 to 0.81) and a trend toward reduced incidence of myocardial infarction (adjusted hazard ratio, =0.69, 95% CI, 0.44 to 1.09) [115]. A prospective observational study of 249 individuals with heart failure with reduced ejection fraction during Ramadan reported that stricter adherence to the daily fasting was associated with stabilization of heart failure symptoms [116].

## 3. TRE Safety

Fasting has been employed for various lengths of times safely for centuries. Fasting for religious reasons is common in two of the most prevalent religions worldwide: Judaism and Islam. Fasting durations employed by individuals following these religious practices is often much more prolonged than for TRE. For example, over the 30 days of Ramadan, individuals fast for anywhere from 10 to 21 h per day depending on their location in the world. In Judaism, personal fasting is undertaken as an act of penance. The most famous fast day of Judaism is Yom Kippur, which consists of a 25-h fast. These religious fasts differ from TRE in that they require total abstinence from food and drink, including water. Water-less fasting has been associated with a state of dehydration [117]. Nonetheless, despite the lack of water, fasting during Ramadan [116,118] and Yom Kippur [119,120] has been found to be relatively safe even for individuals with chronic conditions such as kidney transplant recipients [118], heart failure [116], and diabetes [119,120].

A 2020 systematic review reported that TRE did not cause major adverse events or negatively impact eating disorder symptoms among adults with obesity, metabolic syndrome, or diabetes [40]. Within adults with type 2 diabetes [98,121,122] or pre-diabetes [62], TRE with 15–20 h fasting periods does not cause occurrences of hypoglycemia. In addition, one study reported no impact of TRE on psychological well-being (e.g., depression, anxiety, or stress) [121]. 

Typically, in the process of losing fat mass through a calorie restricted diet, patients can experience a decrease in fat-free or lean mass that contributes to 20–35% of the total weight lost, depending on baseline weight [123]. However, a meta-analysis of ten TRE intervention studies with 241 participants showed no change in fat-free/lean mass (−0.24 kg, 95% CI, −1.15 to 0.67) without significant heterogeneity among the results of the included studies [41]. However, these findings may differ when TRE is performed for longer duration, especially if it results in clinically significant weight loss. In longer duration TRE interventions or in patient populations with baseline low lean mass or frailty, concurrent prescription of regular physical activity (especially resistance exercise training) and high protein intake (1.25–1.50 times the recommended dietary allowance) are recommended to help to reduce the concomitant loss of lean mass [123].

## 4. TRE Feasibility

Typically, self-directed dietary regimens involving caloric restriction and/or macronutrient manipulation require patients to self-monitor and adjust their dietary intake, which is highly burdensome for some individuals [124] and can be inaccurate [125]. Self-monitoring and adjusting dietary intake require estimating calorie content or food volume, weighing each individual ingredient, reading food labels, and/or referencing an electronic nutrition database [125]. Individuals commonly underestimate the caloric value of different food items, by as much as 28% on food items over 500 calories in one study [126]. Weighing each ingredient increases accuracy, but requires the purchase of a food scale, and is tedious, and therefore may not be associated with high adherence. Patients following a ‘free-living’ diet where they purchase their own food and aim to follow a specific prescribed calorie intake or macronutrient ratio will typically need to reference an electronic food database to determine the calorie and macronutrient of different foods. This requires internet access and technological skills that create a barrier in older populations and/or rural areas. An alternative method of reading food labels and self-calculating intake is hampered by reported deficits among adults’ understanding of nutrition labels [127]. A dietary program that provides pre-measured meals and snacks offers a high level of convenience and removes the need for self-monitoring but is associated with significant cost. 

In contrast, TRE is simple to prescribe and follow. It requires minimal instruction and no specialized training or equipment (e.g., a food scale). This in turn results in minimal administrative time and costs for health care practitioners to prescribe TRE. Contrary to other caloric restrictive diets, which can include changes to the macronutrient content of the diet, TRE diets allow participants to continue enjoying the foods they habitually consume. This makes it easier to implement for patients and does not increase food-related costs. There are also free phone applications available to assist patients to track their eating window (e.g., https://www.zerofasting.com/ and https://www.bodyfast.de/en/), but this is not required to be able to follow TRE. The low cost and simplicity of TRE as an intervention reduce common barriers to the adoption and maintenance of lifestyle therapies.

## 5. TRE Adherence and Barriers

A recent systematic review demonstrates that adherence to TRE is high, typically 80–90%, including among individuals with obesity, metabolic syndrome, and diabetes for 4–12 weeks [40]. Other studies have reported even higher rates of adherence, including 98% adherence to five weeks of TRE in one small study of 8 men with prediabetes [62]. Furthermore, dropout rates from TRE studies are lower than in other formats of intermittent fasting (~10 vs. 20%) and much lower than caloric restriction (up to 33%) [128]. While it is likely that the lower dropout rate from studies would translate to greater real-world adherence, there is no research evidence to-date to confirm that TRE is associated with long-term adherence. Individuals following TRE have reported feelings of increased energy, well-being, self-awareness, sleep quality, health-related quality of life and enhanced ability to avoid snacking in the evening [99,111]. These positive qualitative experiences may enhance willingness and motivation to maintain adherence to TRE. 

The primary barriers to longer-term adherence to TRE include incompatibility with family/social life and work schedules [98]. One potential solution suggested by patients who had followed TRE was allowing a more flexible protocol (e.g., weekends off, customized eating window) [128]. Animal data suggest that time-restricted feeding during the week with ad libitum weekend feeding (even with access to high fat and sugar) is similarly effective to continuous (every day) time-restricted feeding for reducing fat mass, improving insulin resistance and normalizing triglyceride levels [129]. To our knowledge, no studies have been published using a weekday only TRE model in humans. However, the positive results from studies reporting 80–90% TRE adherence suggest that, at minimum, one day off from TRE per week is likely to still provide substantial health benefits. It has been suggested that a priori prescription of a planned hedonic goal deviation (e.g., one ‘cheat’ day per week) will enhance long-term adherence to the intervention by enhancing motivation to persist, improving emotional experience, and helping with self-regulation [130]. Therefore, a pre-emptive prescription of TRE for only 5–6 days of the week may be an effective strategy that should be further explored.

A wide variety of TRE protocols with different timing and lengths of the eating window and total durations have been shown to offer metabolic health benefits [39,40]. This accumulated evidence can be used to deduce that there is room for flexibility in TRE protocols to address personal preferences. Accounting for preferences through personalized intervention approaches fosters patient autonomy, enjoyment, and adherence [131,132] A personalized TRE protocol has been suggested in the literature as a strategy to enhance long-term adherence [133]. Potential personalization modifications with evidence for health benefits [39] include a) personalizing the eating window time of day as long as it ends ≥3 h prior to bedtime, and if possible, at or prior to 8 pm; b) personalizing the eating window length to 4–10 h (8 being most common); c) performing TRE on 5–7 continuous days per week. Future research is needed to evaluate the effect of combining two or more of these components of personalization to determine the breadth of flexibility possible for the implementation of TRE as a long-term health behavior. 

## 6. Implications and Future Directions

The potential benefits of TRE as a therapy for primary and secondary breast cancer prevention, and tertiary prevention as it relates to cardiovascular disease in breast cancer survivors are three-fold:TRE directly improves many of the biological and behavioral mechanisms underpinning the development of breast cancer and cardiovascular disease. For example, obesity has been found to have a strong causal relationship with primary and secondary prevention of breast cancer and cardiovascular disease and related mortality. Shared features of the pathogenesis and progression of both conditions that may mediate obesity include oxidative stress and chronic inflammation. Metabolic syndrome, a constellation of metabolic disturbances including hyperglycemia, elevated triglycerides, low HDL, hypertension, and abdominal obesity, is well established to increase the risk of cardiovascular disease and has an emerging strong link to breast cancer. While further research is needed to confirm efficacy on all of these specific outcomes, the available evidence suggests that TRE has promising positive effects on inflammation, oxidative stress, and metabolic health.TRE directly addresses some of the safety and feasibility concerns associated with existing dietary interventions. Mainly, while existing weight loss interventions tend to result in loss of lean mass contributing to 20–35% of total weight loss, TRE has been found to result in decreases in fat mass while sparing lean mass. TRE also removes barriers to participating in dietary interventions, by not requiring tedious calorie counting or use of technology. Preliminary evidence and biological plausibility suggest that personalization of a TRE protocol to an individual’s preferences or lifestyle may enable long-term adherence while still offering health benefits. There are also no costs associated with this intervention. This may be why adherence rates have been reported to be much higher than other dietary interventions, with one study reporting adherence as high as 98%. To-date, most TRE studies have been 8–12 weeks in duration, but due to its simplicity and potential for high adherence, it could be an effective strategy to ameliorate the well-known issue of long-term adherence to health behaviors, especially with allowance of protocol modifications for personal preferences.TRE is safe. Many studies evaluating the practice of fasting during Ramadan and Yom Kippur suggest that, even without consuming water, it can be safe for individuals with chronic conditions such as diabetes and heart failure. It is also likely safe to perform during chemotherapy treatment for breast cancer, based on evidence that longer periods of fasting have been shown to be safe and tolerable, but this requires further research. In addition, no TRE studies have reported the occurrence of major adverse events nor hypoglycemia even among individuals with diabetes. Instead, individuals following TRE have reported positive feelings of increased energy, well-being, and self-awareness.

This accumulating evidence for the potential of TRE to positively impact the development and progression of breast cancer, cardiovascular, and metabolic diseases merits further research, but leaves a number of remaining questions. First, given the short duration of most TRE studies published to-date, interventions with a longer duration and a longer follow-up period are needed to determine the potential for long-term adherence and sustainability of this dietary intervention. Future research should aim to address barriers identified for TRE, such as incompatibility with social or personal life and work schedules. There are potential solutions to overcome these barriers such as personalizing the eating window length and timing and/or incorporating cheat days, that still need to be empirically tested. Lastly, TRE has not been experimentally tested in patients with cancer or cardiovascular disease. Nonetheless, this review described promising observational evidence in these populations and positive experimental evidence on the effects of TRE on biological and behavioral mechanisms underpinning these conditions. In composite, these data suggest that TRE may be an easy and novel lifestyle interventions for the primary and secondary prevention of breast cancer, as well as tertiary prevention as it relates to cardiovascular disease in breast cancer survivors.

## Data Availability

Not applicable.

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
