# Peer review of "Time-Restricted Eating: A Novel and Simple Dietary Intervention for Primary and Secondary Prevention of Breast Cancer and Cardiovascular Disease"

_nutrients, 2021, doi:10.3390/nu13103476_

Round 1

Reviewer 1 Report

The authors provided a comprehensive literature review on time restricted eating (TRE) and its effects in primary and secondary prevention of breast cancer and cardiovascular disease. The manuscript is valuable and clearly written. The adressed topic is interesting and important as it may have clinical implications. I have no major concerns about this review.

Author Response

The authors provided a comprehensive literature review on time restricted eating (TRE) and its effects in primary and secondary prevention of breast cancer and cardiovascular disease. The manuscript is valuable and clearly written. The adressed topic is interesting and important as it may have clinical implications. I have no major concerns about this review.

We thank the reviewer for taking the time to review our manuscript and for the positive comments.

Reviewer 2 Report

This is an interesting and quite exhaustive review of the existing evidence to support that Time Restricted Eating (TRE) may be beneficial for Breast Cancer (BC) and the higher risk of cardiovascular disease (CVD) for patients with BC. The authors conclude that further studies are needed to explore the effect of TRE on the primary, secondary and tertiary prevention of BC and CVD among women with this cancer.  The review includes relevant references supporting the insufficient current evidence supporting beneficial effects of TRE. Thus, it is well justified that TRE should be further investigated, and this review may help to stimulate intervention studies on the effects of TRE on BC, CVA and probably other diseases. See comment to authors)

Major changes.

- There is a problem with the use of primary, secondary and tertiary prevention of breast cancer. Particularly for secondary prevention, defined as early diagnosis and treatment, it is unclear how can help to improve this secondary prevention, for instance in screening programs (how?).

- Thus, the title of the manuscript should be changed avoiding the use of the term secondary prevention, unless well clarified.

- Consequently, the study aim should be also changed, particularly avoiding the term secondary prevention. And it should be questioned instead since the evidence is still insufficient.  How can TRE have “numerous health benefits that are relevant to the primary, secondary, and tertiary prevention of breast cancer”.

- The format for all references should be reviewed in depth, following the requirements of the journal.

Minor changes:

Lines 74-75. No references of support for this sentence.

Lines 75-77. The study aim is unclear, too general. What type of review is intended (meta-analysis? Time period? etc).

Line 81. Metabolic Health not defined.

Line 114 and previous. Fat mass and fat-free mass definitions should be defined briefly.

Lines 133-135 Check TRE effectiveness for secondary prevention. Secondary prevention is not defined and it is hard to believe how TRE may influence early diagnosis and treatment of Breas Cancer. -

2.4. Auxiliary Health Behavior Benefits… Please check, point 2.5 is lacking. 2.6. Cancer-specific Biological Effects

Line 263. “Disruption of circadian rhythms”. Please, define this term of disruption.

Author Response

This is an interesting and quite exhaustive review of the existing evidence to support that Time Restricted Eating (TRE) may be beneficial for Breast Cancer (BC) and the higher risk of cardiovascular disease (CVD) for patients with BC. The authors conclude that further studies are needed to explore the effect of TRE on the primary, secondary and tertiary prevention of BC and CVD among women with this cancer.  The review includes relevant references supporting the insufficient current evidence supporting beneficial effects of TRE. Thus, it is well justified that TRE should be further investigated, and this review may help to stimulate intervention studies on the effects of TRE on BC, CVA and probably other diseases. See comment to authors)

Major changes.

- There is a problem with the use of primary, secondary and tertiary prevention of breast cancer. Particularly for secondary prevention, defined as early diagnosis and treatment, it is unclear how can help to improve this secondary prevention, for instance in screening programs (how?).

We thank the reviewer for this question. Secondary prevention of cancer includes a wide range of approaches, of which early detection/screening is one, but it also encompasses strategies to prevent cancer recurrence and cancer mortality. We did state that this is the aspect of secondary prevention we are discussing in the following sentence in the introduction: “Similar to primary prevention of breast cancer, risk factor targets for secondary prevention as it relates specifically to cancer recurrence, second cancers, and cancer mortality include poor diet[23], physical inactivity[24] and obesity[25,26].” (Line 56 to 57)

 - Thus, the title of the manuscript should be changed avoiding the use of the term secondary prevention, unless well clarified.

As per our response above and below, we have clarified the aspect of secondary prevention of cancer that we are referring to. We also feel that given the title mentions a lifestyle intervention, that it will be obvious that it is not meant to imply screening as a strategy for secondary prevention.

- Consequently, the study aim should be also changed, particularly avoiding the term secondary prevention. And it should be questioned instead since the evidence is still insufficient.  How can TRE have “numerous health benefits that are relevant to the primary, secondary, and tertiary prevention of breast cancer”.

We thank the reviewer for this comment. We have clarified our terminology in the aim to specify the aspect of secondary prevention that we are referring to: “The purpose of this narrative review is to describe relevant peer reviewed literature on the potential for TRE as a therapy for primary prevention of breast cancer, secondary prevention as it relates to recurrence or cancer mortality, as well as tertiary prevention as it relates to cardiovascular disease in breast cancer survivors.” Line 77 to 80

- The format for all references should be reviewed in depth, following the requirements of the journal.

We thank the reviewer for identifying these issues. We have downloaded the reference template from Nutrients and implemented it accordingly.

Minor changes:

Lines 74-75. No references of support for this sentence.

We have included the following reference: O’Connor, S.G.; Boyd, P.; Bailey, C.P.; Shams White,M.M.; Agurs-Collins, T.; Hall, K.; Reedy, J.; Sauter, E.R.; Czajkowski, S.M. Perspective: Time Restricted Eating Compared with Caloric Restriction: Potential Facilitators and Barriers of Long Term Weight Loss Maintenance. Adv. Nutr. 2021, 12, 325–333, doi:10.1093/ADVANCES/NMAA168.    

Lines 75-77. The study aim is unclear, too general. What type of review is intended (meta-analysis? Time period? etc).

We thank the reviewer for this comment. We have clarified this statement to say the following: “The purpose of this narrative review is to describe relevant peer reviewed literature on the potential for TRE as a therapy...” (Bold added just here for emphasis) (Line 77 to 78)

Line 81. Metabolic Health not defined.

We have amended the first sentence about health benefits to provide a definition: “Body mass is an important determinant of metabolic health, which is often signified by the absence of elevated cholesterol, triglycerides, blood pressure, blood sugar and waist circumference.” (line 84 to 85)

Line 114 and previous. Fat mass and fat-free mass definitions should be defined briefly.
We thank the reviewer for this question. The meta-analysis being cited did not provide definitions, but we have added the method of measurement of fat and fat-free mass (i.e., bioelectrical impedance or dual-energy x-ray absorptiometry) which should clarify any concerns. (Line 119 to 122)

Lines 133-135 Check TRE effectiveness for secondary prevention. Secondary prevention is not defined and it is hard to believe how TRE may influence early diagnosis and treatment of Breas Cancer. –

As noted above, we have further clarified our use of the term secondary prevention.

2.4. Auxiliary Health Behavior Benefits… Please check, point 2.5 is lacking. 2.6. Cancer-specific Biological Effects

We thank the reviewer for identifying this error. 2.5 should be cancer-specific biological effects, and heart failure should be 2.6. We have updated this accordingly.

Line 263. “Disruption of circadian rhythms”. Please, define this term of disruption.

Circadian rhythm is defined earlier in the paper as oscillations of physiological rhythms between activity and rest, feeding and fasting, nutrient utilization and storage drive by the natural day/night cycle. The use of the term disruption here has the standard definition of an interruption to a process.

Reviewer 3 Report

This review article describes the potential of time restricted eating as a preventive measure for cardiovascular disease and breast cancer. This topic is of great importance due to the disease burden associated with both cardiovascular diseases and breast cancer and the potential for widespread implementation of the time restricted eating protocol without the requirement for intensive training or extensive nutritional knowledge. Another strength is the description of a dietary intervention that can address multiple health conditions simultaneously.

Please find some minor comments as follows:

  • On line 136, please add missing information: “nutrient deprivation sensitizes cancer cells to the damaging effects of ______”
  • On line 226, please clarify to what relationship you are referring in the following phrase: “but the primary driver of the relationship is volume of visceral fat”.
  • In the TRE Adherence and Feasibility section, it might be worth mentioning some of the adherence challenges, such as social factors, which are described in detail in the paper later.
  • Regarding the phrase “Many studies evaluating the practice of TRE during Ramadan and Yom Kippur” on lines 436-437, based on the definition of TRE in the introduction, it seems that Yom Kippur consisting of one 25-hour fasting period would not be considered TRE. Perhaps consider changing the phrasing in this section to “Many studies evaluating the practice of fasting during Ramadan and Yom Kippur” or something similar to improve clarity.

Author Response

This review article describes the potential of time restricted eating as a preventive measure for cardiovascular disease and breast cancer. This topic is of great importance due to the disease burden associated with both cardiovascular diseases and breast cancer and the potential for widespread implementation of the time restricted eating protocol without the requirement for intensive training or extensive nutritional knowledge. Another strength is the description of a dietary intervention that can address multiple health conditions simultaneously.

Please find some minor comments as follows:

  • On line 136, please add missing information: “nutrient deprivation sensitizes cancer cells to the damaging effects of ______”

We thank the reviewer for catching this error. We have corrected this statement to state “nutrient deprivation sensitizes cancer cells to the damaging effects of chemotherapy”. (Line 142)

  • On line 226, please clarify to what relationship you are referring in the following phrase: “but the primary driver of the relationship is volume of visceral fat”.

We thank the reviewer for this question. We have cut the sentence into two, and now clearly stated what relationship is being discussed: “The primary driver of the relationship between abdominal obesity and poor metabolic health is the volume of visceral fat.” (Line 232 to 234)

  • In the TRE Adherence and Feasibility section, it might be worth mentioning some of the adherence challenges, such as social factors, which are described in detail in the paper later.
    We thank the reviewer for this recommendation and understand the expectation of some mention of barriers in the discussion of feasibility and adherence. To address this disconnect, we combined the adherence paragraph with the barriers paragraph. To accommodate this, we moved the TRE safety paragraph up earlier and then have a section on TRE Feasibility (without adherence) prior to the adherence/barriers section.  (Lines 314 to 403)
  • Regarding the phrase “Many studies evaluating the practice of TRE during Ramadan and Yom Kippur” on lines 436-437, based on the definition of TRE in the introduction, it seems that Yom Kippur consisting of one 25-hour fasting period would not be considered TRE. Perhaps consider changing the phrasing in this section to “Many studies evaluating the practice of fasting during Ramadan and Yom Kippur” or something similar to improve clarity.

We thank the reviewer for this suggestion and agree. We have changed the word TRE to fasting as recommended. (Line 448)